# Depression, anxiety and stress among high school students: A cross-sectional study in an urban municipality of Kathmandu, Nepal

**Anita Karki** [1]☉*, **Bipin Thapa** [2]☉, **Pranil Man Singh Pradhan** [3], **Prem Basel** [3]*

**1** Central Department of Public Health, Institute of Medicine, Tribhuvan University, Kathmandu, Nepal, **2** Department of Child, Adolescent Health and Maternal Care, School of Public Health, Capital Medical University, Beijing, China, **3** Department of Community Medicine, Maharajgunj Medical Campus, Institute of Medicine, Tribhuvan University, Kathmandu, Nepal

☉ These authors contributed equally to this work.
* prembasel11@gmail.com (PB); anita200345@gmail.com (AK)

**Data Availability Statement:** The data that support the findings of descriptive analysis of this study are available in Figshare with the identifier given below: https://doi.org/10.6084/m9.figshare.19203512 The

## Abstract

Depression and anxiety are the most widely recognized mental issues affecting youths. It is extremely important to investigate the burden and associated risk factors of these common mental disorders to combat them. Therefore, this study was undertaken with the aim to estimate the prevalence and identify factors associated with depression, anxiety, and stress among high school students in an urban municipality of Kathmandu, Nepal. A cross-sectional study was conducted among 453 students of five randomly selected high schools in Tokha Municipality of Kathmandu. Previously validated Nepali version of depression, anxiety, and stress scale (DASS-21) was used to assess the level of symptoms of depression, anxiety and stress (DAS). Multivariable logistic regression was carried out to decide statistically significant variables of symptoms of DAS at p-value<0.05. The overall prevalence of DAS was found to be 56.5% (95% CI: 51.8%, 61.1%), 55.6% (95%CI: 50.9%, 60.2%) and 32.9% (95%CI: 28.6%, 37.4%) respectively. In the multivariable model, nuclear family type, students from science or humanities faculty, presence of perceived academic stress, and being electronically bullied were found to be significantly associated with depression. Female sex, having mother with no formal education, students from science or humanities faculty and presence of perceived academic stress were significantly associated with anxiety. Likewise, female sex, currently living without parents, and presence of perceived academic stress were significantly associated with stress. Prevention and control activities such as school-based counseling services focusing to reduce and manage academic stress and electronic bullying are recommended in considering the findings of this research.

## Introduction

Mental disorders contribute to a huge proportion of disease burden across all societies [1]. Among them, depression, anxiety and stress are the leading causes of illness and disability among adolescents [2]. The physical, psychological, and behavioral changes that occur

data that support the findings of inferential analysis of this study are available in Figshare with the identifier given below: https://doi.org/10.6084/m9.figshare.19203491.

**Funding:** The authors received no specific funding for this work.

**Competing interests:** The authors have declared that no competing interests exist.

throughout adolescence predispose them to a variety of mental health issues [3]. Despite this, mental health and mental disorders are largely ignored and not given the same importance as physical health [4].

The existing community-based studies conducted among high school students of various parts of Nepal have reported a wide range of prevalence of symptoms of depression and anxiety. The prevalence of depressive symptoms has been reported to range from 27% to 76% [5–7]. Likewise, the limited studies conducted in Nepal have estimated the proportion of symptoms of anxiety to range from 10% to 57% [7–9]. A nationwide survey conducted in Nepal revealed the prevalence of mental distress among adolescents (13-17years) to be 5.2% [10]. The Global School Health Survey which was a nationwide survey conducted in 2015 reported anxiety among 4.6% of the students [11].

Previous studies have revealed that sex [12–16], staying away from home [17], grade [12, 14, 16], stream of study [18], academic performance and examination related issues [7, 19], cyber bullying [20] were linked with depression. Likewise, sex [8, 21], grade of students and type of school i.e., public or private [8], family type [17], not living with parents, educational level of parents [21] and high educational stress [22] had been the determinants of anxiety as per previous studies.

High school education is an important turning point in the life of academic students in Nepal [23]. As the educational system becomes more specialized and tough in high school, the students become more likely to experience stress at this level. This might put them at risk of developing common mental disorders such as depression, anxiety and stress (DAS). However, there is a paucity of research studies that have assessed DAS among high school students in Nepal.

Exploring the magnitude and risk factors of symptoms of DAS are very crucial to combat the burden of adolescent mental health issues [24]. However, due to limited access to psychological and psychiatric services as well as the significant social stigma associated with mental health issues, anxiety and depression in early adolescence frequently go undiagnosed and untreated, particularly in developing countries such as Nepal. Therefore, this study aimed to estimate the prevalence and identify factors associated with the symptoms of DAS among high school students in an urban municipality of Kathmandu, Nepal.

## Materials and methods

### Study setting, design, and population

This was a cross-sectional survey conducted in randomly selected high schools of Tokha Municipality, Kathmandu District in province no. 3 of Nepal. The data collection period was from 27th August to 11th September 2019. This municipality was formed on 7 December 2014 by merging five previous villages. It has an area of 16.2 sq.km. and comprises 11 wards [25, 26]. The municipality is rich in cultural and ethnic diversity [25]. According to Nepal government records as of 2017, there were total 218,554 students in Tokha municipality in 82 schools. High school students were the study population for this study [26]. In Nepal, high school students comprise of grade 11 and grade 12 students. The high school differs from lower schooling level since the students have the opportunity to enroll in specialized areas such as science, management, humanities and education. High school are also popularly known as 10+2 [27].

### Sample size calculation and sampling technique

Sample size was estimated using the formula for cross-sectional survey [28], $n = Z^2 p(1-p)/e^2$ considering the following assumptions; proportion (p) = 0.24 [12], 95% confidence level, the margin of error of 5%. The estimated proportion used for sample size calculation was based on

proportion of symptoms of anxiety i.e., 24%, as reported by a similar study conducted in Manipur, India [12].

After calculation, the minimum sample size required was 280. After adjusting for design effect of 1.5 to adjust variance from cluster design and assuming non-response rate of 10%, final sample of 467 was calculated. Two-stage cluster sampling was used. A list of all high schools of Tokha municipality was obtained from the education division of the municipality. Out of twelve high schools (8 private schools and 4 public schools), five schools were randomly selected. Within each selected high school further two sections each of grades 11 and 12 were randomly selected. A total of 20 sections were selected, 4 from each selected school, and all the students from the selected sections were included in the study.

## Data collection tools

A structured questionnaire was prepared based on our study objectives which was divided into three sections. The first section included information about socio-demographic, familial and academic characteristics of the students. The second section included two item question to assess socializing among the students which was based on a previous study by Vankim and Nelson [29], two questions to assess bullying among the students based on 2019 Youth Risk Behavior Survey [30] and one item question to assess perceived academic stress. The third section consisted of Depression, Anxiety and Stress Scale (DASS-21) used to assess level of symptoms of depression, anxiety and stress among the students.

DASS-21 is a psychological screening instrument capable of differentiating symptoms of DAS. Depression, anxiety, and stress are three subscales and there are 7 items in each subscale. Each item is scored on a 4-point Likert scale which ranges from 0 i.e., did not apply to me at all to 3 i.e., applied to me very much. Scores for DAS were calculated by summing the scores for the relevant items. and multiplying by two [31]. A previously validated Nepali version of DASS-21 was obtained and used for data collection. Nepali version of the DASS-21 has demonstrated adequate internal consistency and validity. However, in the validation paper, the construct validity of the tool was evaluated against life satisfaction scale and not a systematic diagnostic tool [32]. Reliability for the symptoms of DAS was tested by Cronbach alpha. Cronbach alpha values for DAS were 0.74, 0.77, and 0.74 respectively.

## Data collection procedure and technique

Data was collected after obtaining permission from the municipality's education division as well as individual high schools. The questionnaire was in both English and Nepali language and had been pre-tested among 45 high school students of neighboring municipality. Self-administered anonymous questionnaires were distributed to students in their respective classrooms and requested for participation. An orientation session was conducted for the filling the questionnaire before distribution. Written informed consent was taken from all students prior to data collection whereas additional written parental consent was obtained from students below 18 years of age. One of the investigators herself collected the data from students. After data collection, a session on depression, anxiety, and stress along with the importance of discussing it with the guardians/ teachers and asking for help was conducted.

## Study variables

The study variables are described in Table 1.

**Table 1. Summary of study variables.**

| Variables | Definitions of Variables | Measurements |
|---|---|---|
| **A. Dependent variables** | | |
| Level of depression | Level of symptoms of depression distinguished by DASS-21 scale | Normal (0–9), mild (10–13), moderate depression (14–20), severe depression (21–27) and extremely severe depression (>27)<br>• No Depression (0–9)<br>• Depression (>9) |
| Level of anxiety | Level of symptoms of anxiety distinguished by DASS-21 scale | Normal (0–7), mild (8–9), moderate anxiety (10–14), severe anxiety (15–19) and extremely severe anxiety (>19)<br>• No anxiety (0–7)<br>• Anxiety (>7) |
| Level of stress | Level of symptoms of stress distinguished by DASS-21 scale | Normal (0–14), mild (15–18), moderate stress (19–25), severe stress (26–33) and extremely severe stress (>33)<br>• No stress (0–14)<br>• Stress (>14) |
| **B. Independent variables** | | |
| **Socio-demographic characteristics** | | |
| Age | Age of the student in completed years at the time of the survey | • Below 18<br>• 18 and above |
| Sex | Sex of the participant | • Male<br>• Female<br>• Others |
| Current living status | The current living condition of the student at the time of survey | • With parents (Staying with parents).<br>• Without parents (Staying with relative, staying in hostel, staying with friends, staying with husband/wife, others) |
| Type of family | Type of family based on composition of family members | • Nuclear<br>• Non-nuclear (Joint or Extended) |
| Father's education | The highest level of education attained by the student's father | • No formal education (illiterate, can only read and write in Nepali).<br>• Formal education (Primary, Secondary, Higher secondary, Bachelor's and above) |
| Mother's education | The highest level of education attained by the student's mother | • No formal education (illiterate, can only read and write in Nepali).<br>• Formal education (Primary, Secondary, Higher Secondary, Bachelor's and above) |
| **Academic characteristics** | | |
| Type of school | The type of school where the student was studying at the time of survey | • Public<br>• Private |
| Grade | The current grade of the student at the time of the survey | • Twelve<br>• Eleven |
| Stream/Faculty | The stream or faculty in which student was enrolled at the time of survey | • Humanities/Science<br>• Management |
| Failure in previous examination | Academic record based on the result in the last examination attempted by the student | • Failed<br>• Passed |
| Perceived academic stress | Academic stress as rated by the student for themselves | • Stressed<br>• Not Stressed |
| **Contextual Characteristics** | | |
| Socializing | Socializing status of the students guided by a previous study by Vankim and Nelson | • High<br>• Low |
| Bullied electronically | Bullying status in the past 12 months via any electronic media as reported by the student | • Yes<br>• No |
| Bullied on school property | Bullying status in the past 12 months on school property reported by the student | • Yes<br>• No |

## Data analysis

Compilation of data was done in EpiData 3.1 and then exported to IBM SPSS Statistics version 20 (IBM Corp., Armonk, NY) for cleaning and analysis. Descriptive analysis was performed.

Frequency tables with percentages were generated for categorical variables, while mean and standard deviation (SD) were calculated for continuous variables.

Binary logistic regression was performed to identify associated factors of symptoms of DAS. Firstly, we performed univariate analysis in which each co-variate was modeled separately to determine the odds of DAS. Those variables with p-value <0.15 in univariate analysis were identified as candidate variables for multivariable logistic regression. In multivariable logistic regression, a p-value of < .05 was considered to be statistically significant and strength of association was measured using adjusted odds ratio (AOR) at 95% confidence interval.

Multicollinearity of variables was tested before entering them in the regression analysis. No problem of multicollinearity was seen among the variables (the highest observed VIF was 1.25,1.10 and 1.13 for symptoms of DAS respectively. The goodness of fit of the regression model was tested by the application of the Hosmer and Lemeshow test; the model was found to be a good fit (P >.05).

The regression model was explained by the equation:

$$\text{Log } [Y/ (1-Y)] = b_0 + b_1 X_1 + b_2 X_2 + b_3 X_3 \ldots \ldots b_n X_n + e$$

Where Y is the expected probability for the outcome variable to occur, $b_0$ is the constant/intercept, $b_1$ through $b_n$ are the regression coefficients and the $X_1$ through $X_n$ are distinct independent variables and e is the error term.

### Ethical approval and consent

The study protocol was approved by the Institutional Review Committee (IRC) of the Institute of Medicine, Tribhuvan University (Reference no. 23/ (6–11) 76/077). Approval to conduct this study was also obtained from the education division of Tokha Municipality (Ref: 076/077-23) and respective school authorities. A written informed consent (in the Nepali language) was obtained from the students before the data collection to assure their willingness to participate and no identifiers were listed in the questionnaire to make it anonymous and confidential. Parental consent was obtained for students who were under the age of 18. No incentives were provided.

## Results

### Sociodemographic, academic and contextual characteristics of the students

The research questionnaire was distributed to a sample of 468 high school students, one of whom refused to participate in this study, with a response rate of 99.78%. Responses from 14 students were excluded due to incompleteness. This study presents the analysis on a total of 453 students.

The mean age of the students was 16.99 years (SD = ±1.12), ranging from 14 to 22 years. The proportion of female students (54.1%) was higher than male students (45.9%). Majority of the students were found to be currently living with their parents i.e., 65.8%. Around 70% of the students were from nuclear family. Regarding parent's educational level, majority of the students responded that their father as well as mother had attained secondary level of education i.e., 31.6% and 33.3% respectively.

With regards to academic characteristics, more than two- third of students i.e., 69.5% were from private high schools while the remaining 30.5% were studying in a government or public high school. More than half i.e. (53.4%) of the students studied in grade eleven. About half of the students i.e., 50.6% were from management faculty. Only 3.8% students reported to have failed in the previous examination.

It was noted that about 60% of students perceived themselves to be stressed due to their studies. Most students were low socializing i.e., 60.9%. Around one-tenth students reported being bullied electronically in the past 12 months (10.2%). Similar proportion of students i.e., 10.4% also reported being bullied on school property in the past 12 months (Table 2).

**Table 2. Distribution of the students by socio-demographic, academic and contextual characteristics (n = 453).**

| Characteristics | n | % |
|---|---|---|
| **Age** | | |
| Mean ± SD | | 16.99±1.12 |
| Below 18 | 335 | 74.0 |
| 18 and above | 118 | 26.0 |
| **Sex** | | |
| Female | 245 | 54.1 |
| Male | 208 | 45.9 |
| **Current living status** | | |
| Staying with parents | 298 | 65.8 |
| Staying with relatives | 96 | 21.2 |
| Staying with friends | 19 | 4.2 |
| Staying in hostel | 15 | 3.3 |
| Staying with husband/wife | 1 | 0.2 |
| Staying with brother or sister | 15 | 3.3 |
| Staying alone | 9 | 2.0 |
| **Type of family** | | |
| Nuclear | 319 | 70.4 |
| Joint | 117 | 25.8 |
| Extended | 17 | 3.8 |
| **Father's education** | | |
| Illiterate | 39 | 8.6 |
| Only read and write in Nepali | 64 | 14.1 |
| Primary | 77 | 17.0 |
| Secondary | 143 | 31.6 |
| Higher Secondary Level | 88 | 19.4 |
| Bachelor's and above | 42 | 9.3 |
| **Mother's education** | | |
| Illiterate | 72 | 15.9 |
| Only read and write in Nepali | 90 | 19.9 |
| Primary | 67 | 14.8 |
| Secondary | 151 | 33.3 |
| Higher Secondary Level | 49 | 10.8 |
| Bachelor's and above | 24 | 5.3 |
| **Type of School** | | |
| Private | 315 | 69.5 |
| Public | 138 | 30.5 |
| **Grade** | | |
| Twelve | 211 | 46.6 |
| Eleven | 242 | 53.4 |
| **Stream/Faculty** | | |
| Science | 128 | 28.3 |
| Management | 229 | 50.6 |
| Humanities | 57 | 12.6 |
| Education | 35 | 7.7 |
| Special Law | 4 | 0.9 |
| **Failure in previous exam** | | |
| Yes | 17 | 3.8 |

(*Continued*)

**Table 2.** (Continued)

| Characteristics | n | % |
|---|---|---|
| No | 436 | 96.2 |
| **Perceived academic stress** | | |
| Stressed | 272 | 60.0 |
| Neutral | 93 | 20.5 |
| Not stressed | 88 | 19.4 |
| **Socializing** | | |
| High-socializing | 177 | 39.1 |
| Low-socializing | 276 | 60.9 |
| **Bullied electronically** | | |
| Yes | 46 | 10.2 |
| No | 407 | 89.8 |
| **Bullied on school property** | | |
| Yes | 47 | 10.4 |
| No | 406 | 89.6 |

## Level of symptoms of DAS among the students

The prevalence of symptoms of DAS was found to be 56.5% (51.8%, 61.1%), 55.6% (50.9%, 60.2%) and 32.9% (28.6%, 37.4%) respectively. About a quarter of students showed moderate level of symptoms of depression and anxiety i.e., 25.8% and 24.5% respectively. On the other hand, symptoms of mild stress were most prevalent among the students. i.e., 14.8% (Table 3).

## Factors associated with symptoms of depression

The results from multivariable logistic regression analyses for correlates of symptoms of depression are shown in Table 4. The variables that remain in the final model were age, type of family, father's education, mother's education, type of school, grade, faculty, perceived academic stress, and bullied electronically as these variables had p-value less than 0.15 in the univariate model. In the final model, nuclear family type (AOR: 1.64, 95% CI: 1.06–2.52), students from science/humanities faculty (AOR: 1.58, 95% CI: 1.05–2.40), presence of perceived academic stress (AOR: 1.62, 95% CI: 1.08–2.44) and bullied electronically in past 12 months (AOR: 2.84, 95% CI: 1.34–5.99) were significantly associated with symptoms of depression.

## Factors associated with symptoms of anxiety

The results from multivariable logistic regression analyses for correlates of symptoms of anxiety are shown in Table 5. The variables that remained in the final model were age, sex, mother's

**Table 3. Level of symptoms of DAS among the students (n = 453).**

| Level | Depression | | Anxiety | | Stress | |
|---|---|---|---|---|---|---|
| | n | % | n | % | n | % |
| None | 197 | 43.5 | 201 | 44.4 | 304 | 67.1 |
| Mild | 84 | 18.5 | 39 | 8.6 | 67 | 14.8 |
| Moderate | 117 | 25.8 | 111 | 24.5 | 53 | 11.7 |
| Severe | 39 | 8.6 | 43 | 9.5 | 24 | 5.3 |
| Extremely Severe | 16 | 3.5 | 59 | 13.0 | 5 | 1.1 |
| Overall | 256 | 56.5 | 252 | 55.6 | 149 | 32.9 |

**Table 4. Factors associated with symptoms of depression among the high school students of an urban municipality in Kathmandu (n = 453).**

| Characteristics | Symptoms of depression | | COR (95% CI) | AOR (95%CI) |
|---|---|---|---|---|
| | Yes (%) | No (%) | | |
| **Age** | | | | |
| ≥18 years | 81 (68.6) | 37(31.4) | 2.00 (1.28–3.12) | 1.56(0.96–2.53) |
| <18 years | 175 (52.2) | 160 (47.8) | 1 | 1 |
| **Type of family** | | | | |
| Nuclear | 190 (59.6) | 129 (40.4) | 1.52 (1.01–2.28) | 1.64(1.06–2.52) * |
| Non-nuclear | 66 (49.3) | 68 (50.7) | 1 | 1 |
| **Father's education** | | | | |
| No Formal Education | 68 (66.0) | 35 (34.0) | 1.67 (1.06–2.65) | 1.31(0.79–2.19) |
| Formal Education | 188 (53.7) | 162 (46.3) | 1 | 1 |
| **Mother's education** | | | | |
| No formal education | 106 (65.4) | 56(34.6) | 1.78(1.20–2.65) | 1.46(0.93–2.30) |
| Formal Education | 150 (51.5) | 141 (48.5) | 1 | 1 |
| **Type of school** | | | | |
| Public | 90 (65.2) | 48 (34.8) | 1.68(1.11–2.55) | 1.16(0.73–1.84) |
| Private | 166 (52.7) | 149 (47.3) | 1 | 1 |
| **Grade** | | | | |
| Twelve | 133 (63.0) | 78 (37.0) | 1.65 (1.13–2.40) | 1.22(0.79–1.90) |
| Eleven | 123 (50.8) | 119 (49.2) | 1 | 1 |
| **Stream/Faculty** | | | | |
| Humanities/science | 142(63.4) | 82 (36.6) | 1.75(1.20–2.54) | 1.58(1.05–2.40) * |
| Management | 114(49.8) | 115 (50.2) | 1 | 1 |
| **Perceived Academic Stress** | | | | |
| Stressed | 169 (62.1) | 103 (37.9) | 1.77(1.21–2 .59) | 1.62(1.08–2.44) * |
| Not stressed | 87 (48.1) | 94 (51.9) | 1 | 1 |
| **Bullied electronically** | | | | |
| Yes | 36 (78.3) | 10 (21.7) | 3.06 (1.48–6.33) | 2.84(1.34–5.99) * |
| No | 220 (54.1) | 187 (45.9) | 1 | 1 |

Hosmer and Lemeshow goodness-of-fit test p-value = 0.77

* p< .05

education, stream/ faculty, perceived academic stress, bullied electronically, and bullied on school property (p<0.15). Female sex (AOR: 1.82, 95% CI: 1.23–2.71), no formal education attained by the mother (AOR: 1.63, 95% CI: 1.08–2.47), students from science or humanities faculties (AOR: 1.50, 95% CI: 1.01–2.21), and presence of perceived academic stress (AOR: 1.93, 95% CI: 1.30–2.87), and were significantly associated with symptoms of anxiety.

## Factors associated with symptoms of stress

The results from multivariable logistic regression analyses for main correlates of symptoms of stress are shown in Table 6. The variables that remained in the final model were sex, current living status, grade, stream / faculty, perceived academic stress, bullied electronically and bullied on school property. In the final model, female sex (AOR: 1.54, 95% CI: 1.01–2.34), currently living without parents, (AOR: 1.70, 95% CI: 1.11–2.61), and presence of perceived academic stress (AOR: 2.11, 95% CI: 1.36–3.26) were significantly associated with stress symptoms.

**Table 5. Factors associated with symptoms of anxiety among high school students of an urban municipality in Kathmandu (n = 453).**

| Characteristics | Symptoms of anxiety | | COR (95% CI) | AOR (95%CI) |
|---|---|---|---|---|
| | Yes (%) | No (%) | | |
| **Age** | | | | |
| ≥18 years | 73 (61.9) | 45 (38.1) | 1.41(.92–2.17) | 1.29(0.81–2.04) |
| <18 years | 179 (53.4) | 156 (46.6) | 1 | 1 |
| **Sex** | | | | |
| Female | 150 (61.2) | 95 (38.8) | 1.64 (1.13–2.39) | 1.82(1.23–2.71) * |
| Male | 102 (49.0) | 106 (51.0) | 1 | 1 |
| **Mother's education** | | | | |
| No formal education | 103 (63.6) | 59 (36.4) | 1.66(1.12–2.47) | 1.63(1.08–2.47)* |
| Formal education | 149 (51.2) | 142 (48.8) | 1 | 1 |
| **Stream/Faculty** | | | | |
| Humanities/science | 136(60.7) | 88 (39.3) | 1.51(1.04–2.19) | 1.50(1.01–2.21) * |
| Management | 116(50.7) | 113 (49.3) | 1 | 1 |
| **Perceived academic stress** | | | | |
| Stressed | 170 (62.5) | 102 (37.5) | 2.01(1.37–2.95 ) | 1.93(1.30–2.87) * |
| Not stressed | 82 (45.3) | 99 (54.7) | 1 | 1 |
| **Bullied electronically** | | | | |
| Yes | 32 (69.6) | 14 (30.4) | 1.94 (1.01–3.75) | 1.60(0.80–3.22) |
| No | 220 (54.1) | 187 (45.9) | 1 | 1 |
| **Bullied on school property** | | | | |
| Yes | 31(66.0) | 16 (34.0) | 1.62(0.86–3.06) | 1.36(0.68–2.68) |
| No | 221(54.4) | 185 (45.6) | 1 | 1 |

(Hosmer and Lemeshow goodness-of-fit test p-value = 0.42)

* p< .05

## Discussion

In our study, the prevalence of depressive symptoms among high school students was found to be 56.5%. The existing community-based studies conducted among high school students of various parts of Nepal have reported a wide range of prevalence of depressive symptoms. A study by Gautam et al. reported that more than one quarter i.e., 27% of high school students in a rural setting of Nepal showed depressive symptoms [6]. Similarly, in a study conducted by Bhattarai et. al. in four schools of a metropolitan city in Nepal, it was found that more than 2/5th i.e., 44.2% students exhibited depressive symptoms [5]. Similar proportion of depressive symptoms i.e., 41.6% was also reported by Sharma et. al in a study conducted among adolescent students of public schools of Kathmandu [9]. The prevalence estimated by these studies are lower than the findings of our study [5, 6, 9]. On contrary, a single high school study by Bhandari et al reported depressive symptoms among 76% students [7]. In our study, the proportion of students showing symptoms of anxiety were 55.6%. A study by Sharma et al. revealed that more than half i.e. 56.9% of public high school students showed symptoms of anxiety [9]. Another study by Bhandari et. al, also found out that nearly one out of two students i.e., 46.5% suffered from anxiety [8].These findings are in line with the findings of our study. On contrary, a study by Bhandari reported that only 10% students had mild anxiety [7]. In our study, the prevalence of stress symptoms among students was 32.9%. A study by Sharma et. al reported that more than 1/4th students i.e., 27.5% showed symptoms of stress which corroborates with the findings of our study.

**Table 6. Factors associated with symptoms of stress among the high school students of an urban municipality in Kathmandu (n = 453).**

| Characteristics | Symptoms of stress | | COR (95% CI) | AOR (95%CI) |
|---|---|---|---|---|
| | Yes (%) | No (%) | | |
| **Sex** | | | | |
| Female | 88 (35.9) | 157 (64.1) | 1.35(.91–2.01) | 1.54(1.01–2.34) * |
| Male | 61 (29.3) | 147 (70.7) | 1 | 1 |
| **Current living status** | | | | |
| Without parents | 62 (40.0) | 93(60.0) | 1.62(1.08–2.43) | 1.70(1.11–2.61) * |
| With parents | 87 (29.2) | 211 (70.8) | 1 | 1 |
| **Grade** | | | | |
| Twelve | 79 (37.4) | 132 (62.6) | 1.47 (.99–2.18 ) | 1.23(0.80–1.89) |
| Eleven | 70 (28.9) | 172 (71.1) | 1 | 1 |
| **Stream/Faculty** | | | | |
| Humanities/science | 81(36.2) | 143(63.8) | 1.34(0.91–1.99) | 1.28(0.83–1.96) |
| Management | 68(29.7) | 161 (70.3) | 1 | 1 |
| **Perceived academic stress** | | | | |
| Stressed | 108 (39.7) | 164 (60.3) | 2.25 (1.47–3.44) | 2.11(1.36–3.26) * |
| Not stressed | 41 (22.7) | 140 (77.3) | 1 | 1 |
| **Bullied electronically** | | | | |
| Yes | 22 (47.8) | 24 (52.2) | 2.02 (1.09–3.74) | 1.60(0.83–3.08) |
| No | 127 (31.2) | 280 (68.8) | 1 | 1 |
| **Bullied on school property** | | | | |
| Yes | 22(46.8) | 25 (53.2) | 1.93(1.05–3.55) | 1.48(0.76–2.88) |
| No | 127(31.3) | 279 (68.7) | 1 | 1 |

(Hosmer and Lemeshow p = 0.68)

* p< .05

While the prevalence of symptoms of DAS reported by our study corroborates with the existing literatures in Nepal, it is exceptionally high. One possible explanation for this could be that the data was collected at the beginning of academic session. The students in the eleventh grade were undergoing sudden transition from secondary school life to high school life with regards to new friends, teachers, school environment, and change in daily schedules whereas the students in 12[th] grade were awaiting results of previous board exam. This anticipation and the tremendous pressure faced by 12[th] grade students for tertiary education might have contributed to the high prevalence of symptoms of DAS among 12[th] grade students whereas the higher prevalence of symptoms of DAS among 11[th] grade students could be possibly explained by the inability to cope with the adjustment of sudden transition from secondary to high school life. Moreover, the wide range in prevalence of DAS symptoms among these community-based studies could be attributed to the difference in the setting (rural or urban) and difference in methodology used.

Among South Asian countries, the prevalence of depression reported by our study is in line with the studies conducted in India, and Bangladesh, but slightly higher than one conducted in China and [13, 17, 33, 34]. On contrary, our study has shown higher prevalence of anxiety among students as compared to study conducted in India, Sri Lanka, Vietnam and China [12, 19, 22, 34].The prevalence of symptoms of stress in this study is comparable to the study from Chandigarh but higher than similar study from Manipur, India [12, 17]. Hence, it can be suggested that there is a huge burden of DAS among high school students in South Asia. In context of Nepal, there is no standalone mental health policy. Further, there is inadequate funding

allocated for mental health services along with shortage of qualified mental health professionals. In addition, there is much stigma that surrounds mental illness which acts as a barrier to seek and utilize mental health care services [35]. Due to these reasons, mental health illnesses are likely to remain untreated and continue to persist in the society. This may explain the high prevalence of DAS in our setting.

## Socio-demographic characteristics and association with symptoms of DAS (depression, anxiety and stress)

In current study, it was found that females were more likely to suffer from symptoms of anxiety and stress than their male counterparts. This finding corroborates with the findings from previous studies [19, 21, 36–39]. On the contrary, a study conducted in Dang, Nepal reported that males were 1.5 times more likely to become anxious [8].One possible explanation for this is adolescent stage in girls is marked by hormonal changes as a result of various reproductive events which may have a role in the etiology of anxiety disorders [40]. Furthermore, when compared to boys, girls are more likely to be subjected to stressful situations such as sexual and domestic violence, which may make them more prone to anxiety and stress problems [41].

This study revealed that the students who live in nuclear families were more likely to exhibit depressive symptoms compared to students from joint or extended families. There are more members in a joint family system, which may provide better opportunities for adolescents to share their emotions and issues, hence providing a strong support system that may serve as a protective factor against depression which may be lacking in nuclear families [42]. Moreover, this study also found out that risks of stress symptoms was higher among students who were staying far from their parents. A similar finding was reported by Arif et al., 2019 in Uttar Pradesh, India [43]. One of the possible explanations might be that students who live without their parents may spend a substantial amount of time alone after school, which does not encourage familial intimacy [44]. As a result, they may feel alone and disconnected from their parents [45]. These adolescents may miss out on the opportunity to internalize the support they would otherwise get, leading to increased stress.

In our study, the students who reported no formal mother's education were at greater risk of showing symptoms of anxiety. This was in accordance with other similar studies [38, 46]. The attachment theory provides a robust foundation for understanding how parental behavior affects a child's ability to recognize and manage stressful events throughout their lives [47]. The theory supports that the educated mother plays a stronger parenting role in the development of emotional skills and mental health outcomes in teenagers which might be protective for anxiety.

## Academic characteristics and association with symptoms of DAS

In our study, the students from science or humanities faculties were more likely to have depression and anxiety as compared to management students. This was in line with other studies which showed higher proportion of depressive symptoms among science students. [48]. Generally, science students have to compete more, study longer hours and have a higher level of curriculum difficulty than management students which explains the finding. Likewise, it is believed that the humanities students have a poorer past academic performance in the secondary school, and may have chosen this stream / faculty as a secondary choice [49]. This combined with the uncertainty regarding future work prospects among humanities students may likely explain the higher prevalence of depression among humanities students.

In our study, the students who reported to be stressed due to their studies were more likely to suffer from symptoms of DAS. Several studies have documented similar findings [7, 22]. A possible explanation might be that high school is an important stage in an individual's academic life. However, the inability to meet the expectation of parents, teachers, and oneself in terms of academic performance can lead to overburden of stress [50]. This persistent academic related stress might accelerate the development of mood disorders such as depression, anxiety and stress among the adolescents [51].

### Contextual factors and association with symptoms of DAS

In our study, the risk of depressive symptoms was higher among those students who were bullied via electronic means. Literature suggests that higher the level of cyberbullying/electronic bullying leads to higher the level of depressive symptoms among adolescents [52]. A similar study by Perren et. al demonstrated that depression was significantly associated with cyberbullying even after controlling for traditional forms of bullying [20]. The victims of cyberbullying may experience anonymous verbal or visual threats via electronic means. These repeated incidents can cause the victims to feel powerless which exacerbates the feeling of fear. This can cause significant emotional distress among victims and contribute to development of depressive symptoms [53].

Even though widely utilized in both clinical as well as research setting, DASS scales are screening tools for symptoms of depression, anxiety, and stress. Hence, they cannot be used as a modality for diagnosis. This limitation should be considered when interpreting the findings of this study. Due to its cross-sectional design, this study was unable to establish causal relationship of depression, anxiety, and stress with associated factors. Since the study tools used in this study investigate the habits and activities of the high school students in the past, recall and reporting bias are likely; however, the effect due to potential confounders have been controlled. As Nepal is a culturally diverse country, the findings of only one municipality may not be generalized to the whole country. Therefore, future studies covering a larger population of high school students employing more robust study designs such as interventional studies are recommended to get the real scenario of common mental disorders.

### Conclusion

In conclusion, more than half of the students had depression and anxiety symptoms and nearly one third of the students had stress symptoms. Nuclear family type, students from humanities/science faculty, presence of perceived academic stress, and being bullied electronically were found to be significantly associated with symptoms of depression. Female sex, no formal mother education, students from humanities/science faculty, and presence of perceived academic stress were significantly associated with symptoms of anxiety. Likewise, symptoms of stress were significantly associated with female sex, currently living without parents, and presence of perceived academic stress.

Therefore, prevention and control activities such as school-based counseling services focusing to reduce and manage academic stress and electronic bullying faced by the students are recommended considering findings of this research.

### Supporting information

**S1 File. Questionnaire form used in data collection.**
(PDF)

## Acknowledgments

We are grateful to Tokha municipality for granting permission to conduct the study. Special thank goes to the school management and teachers for their co-ordination during data collection. Lastly, we would like to thank all the study participants for their co-operation and support during the study.

## Author Contributions

**Conceptualization:** Anita Karki, Prem Basel.

**Data curation:** Anita Karki, Bipin Thapa.

**Formal analysis:** Anita Karki, Bipin Thapa.

**Investigation:** Anita Karki.

**Methodology:** Anita Karki, Bipin Thapa, Prem Basel.

**Project administration:** Anita Karki.

**Software:** Anita Karki, Bipin Thapa.

**Supervision:** Prem Basel.

**Visualization:** Bipin Thapa.

**Writing – original draft:** Anita Karki, Bipin Thapa.

**Writing – review & editing:** Anita Karki, Bipin Thapa, Pranil Man Singh Pradhan, Prem Basel.

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
