## [Decision Letter · Decision Letter 0]

15 Mar 2022

PGPH-D-22-00294

Depression, anxiety and stress among high school students: A cross-sectional study in an urban municipality of Kathmandu, Nepal

Dear Dr. Karki,

Thank you for submitting your manuscript to PLOS Global Public Health. After careful consideration, we feel that it has merit but does not fully meet PLOS Global Public Health’s publication criteria as it currently stands. Therefore, we invite you to submit a revised version of the manuscript that addresses the points raised during the review process.

I am less concerned with Reviewer 1's comment 4 about the difference between the humanities and sciences. You do not need to overhaul the paper with respect to this comment, though some response is warranted. 

We look forward to receiving your revised manuscript.

Kind regards,

Khameer Kidia

Academic Editor

Journal Requirements:

1. Please amend your Financial Disclosure statement. If you did not receive any funding for this study, please simply state: “The authors received no specific funding for this work.”

2. Please update your Competing Interests statement. If you have no competing interests to declare, please state: “The authors have declared that no competing interests exist.”

Additional Editor Comments (if provided):

Reviewers' comments:

Reviewer's Responses to Questions

**Comments to the Author**

1. Does this manuscript meet PLOS Global Public Health’s publication criteria? Is the manuscript technically sound, and do the data support the conclusions? The manuscript must describe methodologically and ethically rigorous research with conclusions that are appropriately drawn based on the data presented.

Reviewer #1: Yes

Reviewer #2: Yes

2. Has the statistical analysis been performed appropriately and rigorously?

Reviewer #1: Yes

Reviewer #2: Yes

3. Have the authors made all data underlying the findings in their manuscript fully available (please refer to the Data Availability Statement at the start of the manuscript PDF file)?

Reviewer #1: Yes

Reviewer #2: Yes

4. Is the manuscript presented in an intelligible fashion and written in standard English?

Reviewer #1: Yes

Reviewer #2: Yes

5. Review Comments to the Author

Reviewer #1: This study looking at the prevalence and risk factors for common mental distress among adolescents has its significance as it tries to address a field that has limited studies in this geographical area. I think this manuscript and study could be strengthened.

There are some major concerns that I would like to share with the authors for improvements:

1. The author has written an introduction clearly providing the aim of the study. However, I would like to suggest the author look into recent studies highlighting the current prevalence of proposed outcome variables as many studies have been published since 2015-16. Furthermore, the overall prevalence of mental disorders among Nepalese adolescents provided by the National Mental Health Survey, Nepal-2020 has been reported in the discussion session, but I think it would be better to have it as an introduction as it covers a broader perspective.

2. In methodology, a study from India has been used as a reference for sample size estimation. I wonder why the author didn’t look for any Nepalese past prevalence. Moreover, the reference study has provided the proportion for depression, anxiety, and stress while only the proportion of anxiety has been used by the author for sample estimation. Was the proportion of anxiety selected purposively to obtain optimal sample size or was there any other reason to ignore the provided prevalence of depression and stress? I suggest the author can aim to make the sample estimation section more explanatory and clear.

3. Is there any specific reason that the authors set the level of significance at 15% for bivariate analysis while the AOR has been interpreted at a 5% level of significance?

4. Humanities and science are very diverse disciplines. Is there any specific reason that the author has grouped science and humanities while management as its counterpart? Moreover, other disciplines are mentioned in Table 2 which are merged during cross-tabulation without clear explanation. Furthermore, in the discussion author has tried to emphasize the complexity of science as a discipline but this interpretation seems misleading as humanities have also been accommodated in the same group in the current manuscript.

5. In the discussion section, it is nice that the author has mentioned the national prevalence of mental disorders among adolescents based on the National Mental Health Survey 2019-2020. However, the study has been referenced as a study from 2018 but the factsheet of this national survey was published in 2020 so the author might need to revise the reference accordingly.

6. I recommended that the author revisit their references focusing on studies based on adolescents and/or high school students. Medical/health science students might have more diverse characteristics than high school students. The author could shape the discussion by revisiting recent studies which have focused on depression among higher secondary school adolescents in Nepal and the Asian region.

7. In reference, some of the references are not attainable/assessable for instance current reference 29. Cross verification of the reference is required

Reviewer #2: This manuscript presents results from a cross-sectional survey of HS students in Kathmandu. Given the dearth of local epidemiological data, especially among young populations, this is an important study.

1. Given the variation in how “high school” is defined across the world, it will be helpful to describe it. It appears that 11th and 12th grades are included.

2. Please include that the validation paper [Ref 31] conducted construct validity against a life satisfaction scale. As such, the scale wasn’t validated against a systematic diagnostic instrument (e.g., a gold standard like SCID).

3. Perhaps relatedly, the overall prevalence is very high. Globally, community samples show 5% prevalence of depression but this study found >50% rate. While such rate may be possible, it is unusually high. Indeed, many community-based studies find a very wide range of prevalence. Additional context will be helpful to interpret these high rates. How close were upcoming examinations? Any other stressors that may have increased overall mood and anxiety? How common is it for HS students to be asked about these issues when they are in the classroom?

4. Overall, the language needs to be edited from “depression” and “anxiety” to “symptoms of depression” or “depressive symptoms” to clarify that the survey instrument is not diagnostic and is picking up presence of symptoms.

6. PLOS authors have the option to publish the peer review history of their article (what does this mean?). If published, this will include your full peer review and any attached files.

**Do you want your identity to be public for this peer review?** For information about this choice, including consent withdrawal, please see our Privacy Policy.

Reviewer #1: No

Reviewer #2: No

---

## [Editor Report · Decision Letter 1]

4 May 2022

Depression, anxiety and stress among high school students: A cross-sectional study in an urban municipality of Kathmandu, Nepal

PGPH-D-22-00294R1

Dear Ms. Karki,

We are pleased to inform you that your manuscript 'Depression, anxiety and stress among high school students: A cross-sectional study in an urban municipality of Kathmandu, Nepal' has been provisionally accepted for publication in PLOS Global Public Health.

Best regards,

Khameer Kidia

Academic Editor